# Postoperative Pain Following Root Canal Instrumentation Using ProTaper Next or Reciproc in Asymptomatic Molars: A Randomized Controlled Single-Blind Clinical Trial

**DOI:** 10.3390/jcm11133816

**Published:** 2022-07-01

**Authors:** Patrícia Santos Oliveira, Meire Coelho Ferreira, Natália Gomes Nascimento Paula, Alessandro Dourado Loguercio, Renata Grazziotin-Soares, Gisele Rodrigues da Silva, Helena Cristina Santos da Mata, José Bauer, Ceci Nunes Carvalho

**Affiliations:** 1Postgraduate Program of Dentistry, CEUMA University, São Luís 65065-470, Brazil; patriciaso_cd@hotmail.com (P.S.O.); meirecofe@hotmail.com (M.C.F.); cd_ngomes@hotmail.com (N.G.N.P.); 2Department of Restorative Dentistry, School of Dentistry, State University of Ponta Grossa, Ponta Grossa 84010-330, Brazil; aloguercio@hotmail.com; 3College of Dentistry, University of Saskatchewan, Saskatoon, SK S7N5B2, Canada; regrazziotin@gmail.com; 4Department of Operative Dentistry, Federal University of Uberlândia (UFU), Uberlândia 38408-100, Brazil; giselerosilva@yahoo.com.br; 5Private Practice, Da Mata Odontologia, São Paulo 04020-040, Brazil; helenamata@live.com; 6Dentistry Biomaterials Laboratory (Biomma), School of Dentistry, Federal University of Maranhão, São Luís 65080-805, Brazil; bauer@ufma.br

**Keywords:** asymptomatic molars, endodontic treatment, postoperative pain

## Abstract

Aim: The development of postoperative pain following root canal instrumentation may impair patient’s comfort and undermine their trust in the dentist. This study assessed the effect of root canal instrumentation techniques (rotary (PTN; ProTaper Next^®^) and reciprocating (R; Reciproc^®^)) on the postoperative pain intensity (primary outcome) and tenderness on biting (secondary outcome) of patients’ asymptomatic molars. Methodology: This study protocol was registered with ReBec-WHO (U1111-1182-2800). From a pool of 112 patients evaluated for eligibility (healthy adults (≤18 years old)), with a single asymptomatic molar (maxillary or mandibular) indicated for root canal treatment, diagnosed with asymptomatic irreversible pulpitis (including chronic hyperplastic pulpitis), 75 were randomly allocated in similar proportions to receive the intervention (two-appointment root canal therapy) in either the PTN or R group. The allocated procedures were performed using standardized protocols. Participants (blinded to the instrumentation technique) rated their pain intensity at 6, 12 and 24 h and from day 2 to day 7 following the root canal instrumentation appointment using a VAS and an NRS; the ibuprofen tablets taken and the presence of tenderness on biting were recorded. The instrumentation time was registered. Univariate and multivariate statistics measured the effect of independent variables on the outcomes. Results: From the 75 patients allocated, 8 patients (4 from each group) were lost; in total, 33 patients were analyzed in the PTN group and 34 in the R group. The frequencies of postoperative pain (*p* > 0.05) and tenderness on biting (*p* > 0.05) were similar between groups. The medication intake (mean of 1.31 tablets) and the time of instrumentation (approximately 11 min) were similar between groups. Conclusion: ProTaper Next and Reciproc^®^ caused a slight risk of tenderness on biting and contributed to similar self-reported postoperative pain (low intensity) up to 7 days following root canal shaping.

## 1. Introduction

Postoperative pain is an undesirable symptom that dental patients may experience after root canal instrumentation, and it can occur as high as 58% of the times [1,2,3,4]. Interappointment pain may cause patients’ sleep disturbance, impair everyday life activities and undermine the trust in the dentist [5,6]. Either tenderness on biting or severe pain can occur within a few hours or days after the dental appointment, as a result of an acute inflammatory response in the periapical tissues [7].

Many factors influence the development of postoperative pain following root canal instrumentation, such as the presence of pre-operative pain, deficient root canal technique (poor disinfection, unlocalized canals, lack of irrigant solution or irrigant solution extrusion) [1], higher concentrations of sodium hypochlorite [8], tooth restoration in hyperocclusion, [1] etc.

During the instrumentation phase, the extrusion of debris and/or microorganisms beyond the tooth apical foramen [9] and the incorrect establishment of the working length, inducing mechanical trauma to the periapical tissues [10], may cause postoperative pain. Even when the clinician performs an optimal root canal therapy and the instruments do not impinge the periapical tissues, debris can be extruded trough the foramen, resulting in an inflammation process [11,12].

To minimize postoperative complications, an instrumentation technique should avoid injuries to the perirradicular tissues and smoothly reach the apical third reducing the debris extrusion. The literature on postoperative pain has showed that both rotary or reciprocating instrumentation kinematics were able to produce positive or negative outcomes. For instance, the rotary technique was associated with reduced debris extrusion in vitro [13] and was clinically correlated to shorter and less intense postoperative pain when compared with reciprocating [14]. The authors claimed that the ProTaper Next^®^ system, because of its offset design and its “wave” movement, would improve debris removal and flexibility in the working part of the file, effortlessly reaching the apical third [2,15]. Contrastingly, in other studies, the reciprocating technique was also linked with lower amounts of debris extrusion [16,17,18], more predictability in relation to the intake of analgesics [9] and reduced preparation times (in the case of Reciproc^®^) due to being a single-file technique [19,20]. In addition to that, some trials have found similar outcomes (postoperative pain and intake of analgesics) for both techniques [2,3].

Recognizing these divergent findings, the reciprocating technique could be more convenient for the patient, having a reduced dental-chair time and, consequently, acting in favor of patient’s comfort. This randomized clinical trial aimed to assess the effect of root canal instrumentation techniques (ProTaper Next^®^ and Reciproc^®^) on the postoperative pain intensity in patients’ asymptomatic molars (primary outcome) and tenderness on biting (secondary outcome) considering the following variables: patients’ gender, oral hygiene (good < moderate < poor) and tooth location (mandibular/maxillary), as well as the intake of analgesics during the follow up period and time of root canal instrumentation.

## 2. Materials and Methods

### 2.1. Study Design, Setting and Sampling

The protocol of this parallel-group, single blind, randomized controlled clinical trial and the informed consent form were approved by the Research Ethics Committee of the university where the study was created (#1.423.099/2015). The study protocol was registered on www.who.int/ictrp/network/rebec/en/ (accessed on 26 December 2016) (ReBec-WHO: U1111-1182-2800). Study reporting was grounded on the Consolidated Standards of Reporting Trials (CONSORT, 2010) guidelines [21], and it was written according to the Preferred Reporting Items for Randomized Trials in Endodontics (PRIRATE) 2020 guidelines [22]. Patients enrolled in this study came from the dental clinic located at Dental Specialty Civic Center (Imperatriz, MA, Brazil) between February/2016 and February/2022. Experiments were undertaken with the understanding and written consent of each subject and according to the Declaration of Helsinki (2008) (www.wma.net (accessed on 26 December 2016)).

### 2.2. Sample Size Calculation

Sample calculation was performed to compare postoperative-pain means as a superiority trial and to verify the null hypothesis of no difference between groups. The Sealedenvelope^TM^ (Sealed Envelope Ltd., London, UK) online calculator was utilized, considering a confidence level of 95%, a power of 80%, mean outcomes in the control (PTN) and experimental (R) groups of 0.59 and 2.18, respectively, and a standard deviation of outcome of 2.3 [14]. A minimum sample of 33 patients was calculated for each group in order to achieve a 95% confidence of a true difference between the groups.

### 2.3. Eligibility Criteria, Patient Selection and Allocation

Patients were selected from a pool of 112 patients, with a noncontributory medical history, who presented at the dental clinic for examination during the study period. Eligible patients included were systemically healthy adults (≤18 years old), with one single asymptomatic molar (maxillary or mandibular) indicated for root canal treatment, pulp/periapical diagnosed with asymptomatic irreversible pulpitis (including chronic hyperplastic pulpitis) and normal apical tissues. Diagnosis was based on chief complain, visual inspection of the tooth, radiographic exam, periodontal examination (probing, palpation and percussion) and pulp sensibility test with cold spray. The exclusion criteria were pregnancy, systemic disease with contraindication for root canal treatment, having taken an analgesic or anti-inflammatory agent prior to treatment, preoperative pain of any origin, internal and external resorption, periodontal disease, teeth undergoing orthodontic treatment, teeth with anatomic abnormalities and/or severe root canal curvature.

Randomization was conducted using the Sealed Envelope^TM^ website (www.sealedenvelope.com (accessed on 1 March 2016)). Block randomization was performed in order to have a list of identical numbers of patients in the control and experimental groups. To ensure allocation concealment and to control selection bias, the assistant (allocator) hid the block size from the operator and used mixed block sizes. Once the patient entered the clinic and the operator verified the fulfillment of the inclusion criteria, the list was then checked by the assistant to verify in which group the patient would be allocated. As patients were the outcome assessors, they were blind to the assigned groups. Patient and tooth-related factors (age, gender, oral hygiene status (good < moderate < poor), tooth location and number of canals) were registered.

### 2.4. Root Canal Treatment Procedures

A single operator (P.O.S) experienced in both instrumentation techniques used in the study (5 years of practice limited to endodontics and 1 year of usage of PTN and R instruments) performed all the root canal treatments, in two appointments each. After the administration of local anesthesia (2% lidocaine with 1.100,000 epinephrine), the affected tooth was isolated with a rubber dam. Following pulp chamber access and cleaning, the canals were explored with #10 and #15 K files. Working length was determined with a foramen locator (using the “0.5 mark” on display) coupled to a VDW Gold Reciproc motor (VDW, Munich, Germany). The tooth then received one of the following instrumentation techniques:Rotary instrumentation group (PTN): Speed was set at 300 rpm and torque at 2 N/cm. X1 was initially used for cervical preparation, and X1 and X2 reached the WL, using in-and-out movements. X3 was used as a master apical file in narrow canals and X4 in large canals.Reciprocating instrumentation group (R): R was operated in “Reciproc All” mode. R25 was used in narrow canals, and R40 was used in large canals. Three in-and-out motions were applied with stroke lengths not exceeding 3 mm in the cervical, middle and apical thirds until attaining the established WL.

For both groups, the instruments were carefully cleaned to remove debris. The operator used a sterilized gauze around the instrument shift, after 3 in-and-out movements (pecking motions). Irrigation was performed with 15 mL of 2.5% sodium hypochlorite for each canal. Final irrigation was performed with 2 mL of 17% EDTA and 2 mL of 2.5% sodium hypochlorite. Apical patency was maintained with a #15 K file beyond the working length [23]. A person not involved in the research study (operator’s assistant) used a timer to record the whole period since the operator firstly inserted the instrument into the canal until the last withdrawal of the instrument(s). This time was defined as “the instrumentation time”—representing the time from the beginning to the end of root canal preparation.

Canals were dried using aspiration with a capillary tip and absorbent paper points. No intracanal medication was used [2,14,24]. The access cavity was temporarily sealed with glass ionomer cement (a Dycal layer was placed underneath to prevent the glass ionomer from obstructing the canal), and occlusion was verified and adjusted if necessary.

### 2.5. Postoperative Pain, Intake of Analgesic and Tenderness on Biting Assessments

Each patient received a pain diary to record their pain at the following time points: 6, 12 and 24 h and from day 2 to day 7 following the procedure. The pain diary consisted of two scales to record pain intensity and two questions, one about analgesic intake (number of 400 mg tablets) and the other about tenderness on biting (yes-or-no question). First, patients pointed a mark in a numerical rating scale (NRS) that was a ten-point scale (10 cm) measuring 0 for no pain, 1–2 for mild pain, 3–4 for moderate pain, 5–7 for considerable pain and 8–10 for severe pain. Second, patients pointed a mark in a visual analogue scale (VAS) that was a 10 cm horizontal line with only two scores at the ends: 0, for no pain and 10 for severe pain. The distance from the patient’s mark to the 0 value was considered as a numerical value for the pain intensity. The patients were prescribed with medication following treatment; they were recommended to take ibuprofen only if they felt moderate-to-severe pain (400 mg—every 6 h, while having pain) [25]. This recommendation was based on the non-contributory medical history of the included patients. If the pain did not alleviate, patients were instructed to contact the operator. For tenderness on biting, patients were given 3 × 2 cm latex devices to bite to be able to report the existence of this symptom. Patients were instructed to position the latex device over the treated tooth and bite it for approximately 3 s (up to three times). Participants returned their pain diaries at their second appointment, scheduled 7 days after the first visit.

### 2.6. Statistical Analysis

Statistical Package for Social Sciences (IBM SPSS version 21.0; IBM Corporation, Armonk, NY, USA) was used for the analysis, and statistical significance was set at *p* < 0.05. Baseline demographic and clinical features (age, gender, oral hygiene status, tooth location and number of canals), as well as time of instrumentation and number of analgesics taken, were compared between groups. A multiple intra-group analysis (PTN group and R group) of postoperative pain for the evaluated time intervals (6, 12 and 24 h and from day 2 and to day 7) were assessed using the Friedman test (*p* < 0.05). The Wilcoxon test was applied every two time intervals, with significance corrected using Bonferroni (*p* < 0.006). The Mann–Whitney test was used to compare the postoperative pain intensity between the groups and to compare the number of analgesic tablets ingested by the patients (*p* < *0*.05). The effect size was calculated for standardized differences between the means of postoperative pain intensity in the groups using Cohen’s d^26^. The ANOVA with repeated measures investigated if the ingestion of analgesics (using the time intervals when patients took more analgesics: 6 h, 12 h and 24 h) had an interaction with postoperative pain, considering each group (instrumentation technique).

The number of events (postoperative pain and tenderness on biting) was compared between groups (using the chi-squared test) and assessed as absolute and relative risks for the outcomes tested. Poisson regression used the postoperative pain (absent/present) at the 12 h time interval (time interval that had the highest postoperative pain recorded by the NRS) as the dependent variable and gender, oral hygiene status (good < moderate < poor), tooth location (mandibular/maxillary) and instrumentation technique (PTN/R) as independent variables. All the associations with *p* < 0.20 in the non-adjusted analysis were included in the adjusted analysis.

## 3. Results

In total, 112 patients were evaluated for eligibility. A total of 37 were excluded from the sample because they did not meet the inclusion criteria, and 75 (from 18 to 66 years old) were included in the trial, with 38 being allocated and receiving the intervention in the R group and 37 being allocated and receiving the intervention in the PTN group. Eight patients (four from each group) were lost, because they did not show up for the second appointment (consequently, they did not return the pain diary), even after the researchers attempted to contact them by text message and phone call. Therefore, 34 patients were analyzed in the R group, and 33 patients were analyzed in the PTN group. The PRIRATE 2020 flow diagram shows the number of patients included in the clinical trial (Figure 1). Table 1 describes patient and tooth-related factors for the two groups: rotary (PTN) and reciprocating (R). There were no statistic significant differences between groups for the time of instrumentation (*p* = 0.536) and intake of analgesics (*p* = 0.988) (Table 1).

Table 2 shows the mean of postoperative pain intensity according to the VAS and NRS scales for each group.

The VAS showed that postoperative pain intensity was highly significant in intra-group comparisons for the evaluated time intervals. The pain peak was up to 12 h. The reduction in pain intensity for the PTN group began 6 h after the procedure, showing a tendency to decrease, but with a statistically significant reduction only between day 4 and day 5; for the R group, pain reduction began 6 h after the procedure and continued for the entire analyzed time but without statistical significance. The reduction was faster for the R group. The VAS indicated no significant differences between groups for all evaluated time intervals, but a moderate magnitude of effect was observed between groups for the periods from 24 h to day 5.

The NRS also showed that the pain peak was up to 12 h. The NRS measured similar postoperative pain intensity values when compared to the VAS scale for the PTN group. Contrastingly, the NRS showed a statistically significant reduction in pain intensity between day 3 and day 4 for the R group. The NRS indicated no statistically significant differences between groups for all evaluated time intervals, but a moderate magnitude of effect was observed between groups for the periods from 24 h to day 4. 

There were no significant differences in the means of the analgesics taken between groups (*p* = 0.988). The overall mean of the analgesics taken by the patients was 1.31 (±2.82) tablets. A total of 16 patients, out of 20 who needed analgesics, took the medication only up to day 2 after the dental procedure. The intake of analgesics by the patients at 6 h (*p* = 0.112), 12 h (*p* = 0.648) and at 24 h (*p* = 0.785) had no influence on their postoperative pain, regardless of the root canal instrumentation technique.

The frequencies of postoperative pain (primary outcome) and tenderness on biting (secondary outcome) were similar between groups. The relative risk for both outcomes showed that one instrumentation technique was not worse than the other in influencing postoperative pain intensity (Table 3).

A patient’s moderate oral hygiene was found to be a factor of protection against postoperative pain in both the univariate (RR = 0.49; 95% CI = 0.27–0.87) and multivariate (RR = 0.48; 95% CI = 0.27–0.85) models. The tooth location did not explain the primary outcome in the multivariate model (Table 4).

## 4. Discussion

This randomized clinical trial showed that the ProTaper Next^®^ (PTN) or Reciproc^®^ (R) endodontic instruments caused the same postoperative pain intensity in patients, up to one week following root canal instrumentation. Since there were no statistical differences between the PTN and R instruments (in either univariate or multivariate analyses), this trial found that both systems may be clinically used with the same results in relation to postoperative pain to prepare patients’ root canals of molars. Molars have been previously associated with greater susceptibility to postoperative pain amongst all tooth types, because of their complex anatomy that renders debridement more difficult, increasing the risk of postoperative complications [5]. 

The generalizability of the study results has to be exercised with caution, because this trial investigated postoperative pain only after root canal instrumentation, which was performed in the first appointment, and not after root canal obturation. Therefore, the applicability of these findings is meaningful in regard to maintaining patients without pain (they were not in pain before seeing the dentist), ensuring their well-being and helping to build a trustworthy patient–dentist relationship, which is important for future appointments.

The patients in this study experienced low pain intensity, around 1 or 2 points in both the visual analog and numerical scales. Overall, the peak of pain was up to 12 h after the procedure, and it reduced as the time passed for the patients of both groups. One important finding was that postoperative pain reduced in intensity faster for patients from the reciprocating (R) group. The reduction was noticed around the third day after root canal instrumentation (significant time point for time reduction measured by the NRS), while for patients from the rotary (PTN) group, the significant reduction in pain was observed on the fourth and fifth days after the procedure (both scales).

The time of instrumentation was measured to monitor if PTN group treatment took longer since it is based on a multifile system, while R group treatment is based on a single-file system. Laboratory studies have demonstrated that all canal preparation techniques are associated with dentin debris extrusion from the root canal system [26,27,28]. Based on this, it can be hypothesized that a longer instrumentation time could cause relatively more injury to the periapical tissues, less amounts of debris extrusion and, consequently, more postoperative pain. Even though the working time was similar between groups, we assume that the observed faster reduction in postoperative pain in the R group could be a result of less manipulation of tissues at the working-length level (meaning, few touches of the periapical tissue, less mechanical trauma and, consequently, less additional inflammation) and, perhaps, less debris extrusion. Regarding the lack of differences in the working time between groups, we think that it might have happened due to our choice of not using a glide path motor-driven instrument before using the R instrument, resulting in more challenges (and time) in reaching the established working length. 

This result differs from a previous clinical trial that showed more intensity and duration of postoperative pain in patients who received root canal treatment with reciprocating files compared with those who were treated with rotary files [14]. Many are the explanations for this discrepancy, i.e., different types of teeth (those authors also included premolars in the sample), different population (pain is multifactorial and influenced by factors inherent to patients), different irrigation solution (they used 2% chlorhexidine in their study), different time evaluation (they measured pain after instrumentation and obturation) and, at last, different types of systems (they used ProTaper Universal (rotary) and WaveOne (reciprocating)). 

Having knowledge of the number of medication tablets used by the patient after root canal instrumentation reinforces the dentist’s awareness of the presence and intensity of postoperative pain. Although no statistical differences were found in the means of analgesics taken between patients from the PTN and R groups (univariate analysis), a numerical difference was found in the absolute number of analgesics taken—patients from the PTN groups took 38 analgesics in total, while patients from the R group took 32 in total. 

On average, 16 patients (out of 20 who needed analgesics) took 1.31 tablets of Ibuprofen (400 mg) in the first 2 days following root canal instrumentation. Other clinical trials have also used Ibuprofen as the first choice to treat postoperative pain and reported that the first 48–72 h after the procedure were the most critical for the patients [8,29]. Other authors have found that 83% of the patients had no pain after 2 days (48 h) following root canal retreatment performed with rotary (MTwo) or reciprocating instruments (Reciproc) [29].

This study is novel because it compared PTN and R and only included patients without pre-operative pain, since having pain ahead of the procedure is a predictor of the development of postoperative pain [5,8,30,31]. In a longitudinal, prospective study, the authors previously showed that the prevalence of post instrumentation pain in patients undergoing two-visit root canal treatment was high (64.7% experienced some level of pain on either day 1 or 2 after the procedure), and this symptom was significantly affected by preoperative pain, but less than 10% of patients experienced severe pain (that was classified as 4 or 5 points on the VAS) [5]. In our study, although this feature (pre-operative pain) could be distributed at random, we decided to exclude patients with pre-procedure sensitivity/pain/discomfort to avoid the overestimation of postoperative pain. A limitation of this clinical trial was the inherent imprecision of the measurement of the pain outcome—a multifaceted and subjective assessment [32].

The standardization of the study protocols aimed to minimize or even eliminate intra-operative variables in the outcomes. As this trial had a single clinician operator, many steps of the root canal treatment that could influence the postoperative pain were controlled well, such as impact and velocity of pecking motions, style of performance of apical patency and pressure on irrigation, i.e., factors that are peculiar to each operator. Another strength of this study was the use of two types of instruments (the VAS and the NRS) to measure pain. It is believed that postoperative pain intensity can be more precisely measured when more than one scale is used—since comparing the relationships between each intensity scale and a derived composite represents the “best possible” assessment of a self-reported construct [33,34]. Despite this, the authors of this study are aware that the variability in pain thresholds amongst different persons may affect the responses regardless of the type and/or number of scales used [35]^.^

Another positive point to be highlighted in this study is the use of a biting latex device to assess tenderness on biting. This study was pioneer in using this type of device and found high acceptance by the patients, since they easily comprehended the difference between tenderness on biting (manifested as a provoked sensation) and postoperative pain. The use of a latex device to reproduce symptoms of tenderness was our choice because it is more comfortable for the patient, and it presents a higher level of sensitivity than wood sticks or cotton rolls. The results showed that there was no increased risk of tenderness on biting for one of the instrumentation techniques over the other. For both groups, there was a (slight) possibility of tenderness (four patients in the PTN group and nine patients in the R group).

Implications for clinical practice include the possibility to use either the ProTaper Next^®^ or Reciproc^®^ system to instrument root canals with the expectation that patients experience low postoperative pain intensity and that they take only a few ibuprofen tablets (fewer for Reciproc^®^). These two instrumentation systems/techniques allow patients to bite and chew in a relatively normal way after the procedure. In addition to that, the dentist has the option to use Reciproc^®^ with the expectation that the postoperative pain decreases faster.

## 5. Conclusions

Postoperative pain was of low intensity for patients who received root canal treatment with either the ProTaper Next^®^ or Reciproc^®^ instrument. Both root canal instrumentation techniques caused some risk of tenderness on biting. The patients’ intake of medication was similar regardless of the instrumentation technique. The operator needed a similar amount of time (approximately 11 min) to complete root canal instrumentation using both systems.

## Figures and Tables

**Figure 1 jcm-11-03816-f001:**
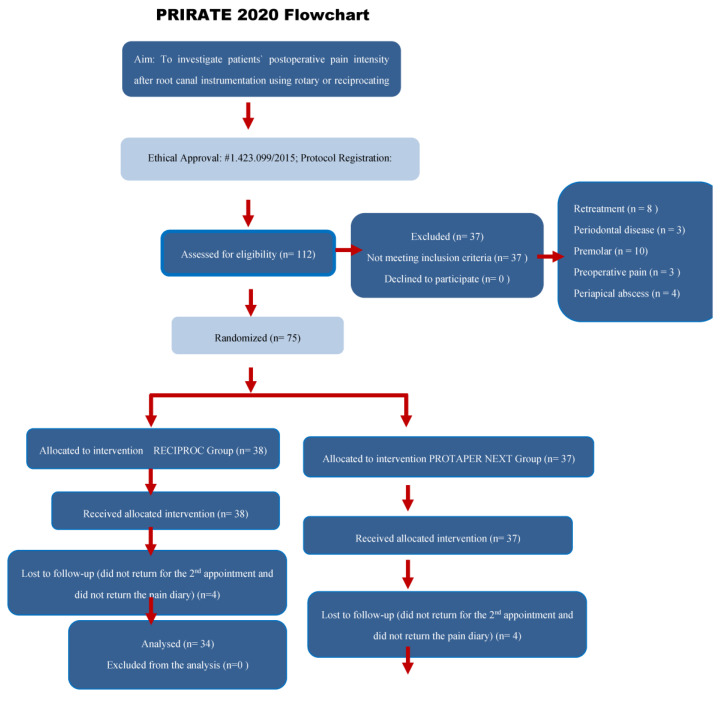
PRIRATE 2020 flow diagram showing the number of patients included in the clinical trial and the main details of the study [22].

**Table 1 jcm-11-03816-t001:** Baseline demographic and clinical features (patient and tooth-related factors) of patients in the study groups.

		Rotary (PTN)	Reciprocating (R)	*p*-Value
		*n* (%)	*n* (%)	
Patient’s gender	Male	16 (48.5)	11 (32.4)	0.178 *
	Female	17 (51.5)	23 (67.6)	
Tooth location	Maxillary Molar	12 (36.4)	9 (26.5)	0.383 *
	Mandibular Molar	21 (63.6)	25 (73.5)	
Number of canals	2	3 (9.1)	1 (2.9)	0.764 **
	3	24 (72.7)	30 (88.2)	
	4	6 (18.2)	3 (8.8)	
		Mean (SD)	Mean (SD)	
Patient’s age		31.12 ± 6.59	32.09 ± 11.10	0.559 ^§^
Instrumentation time (min)		13.21 ± 9.13	10.62 ± 4.30	0.536 ^§^
Number of analgesics taken		1.30 ± 2.88	1.32 ± 2.80	0.988 ^§^

SD—standard deviation; *—chi-squared of Pearson; **—chi-squared of linear trend; ^§^—Mann–Whitney test; PTN—ProTaper Next; R—Reciproc.

**Table 2 jcm-11-03816-t002:** Mean, standard deviation (±SD) and 95% confidence intervals (CI) of postoperative pain intensity, measured by the VAS (0–10 cm) and the NRS (0–10 cm), at the assessed time intervals after root canal instrumentation for both groups.

Scale	Group	6 h	12 h	24 h	Day 2	Day 3	Day 4	Day 5	Day 6	Day 7	*p* *
VAS	Rotary (PTN) ^§^	1.41 ± 2.35 ^a^ (0.52–2.31)	1.28 ± 2.05 ^a^ (0.50–2.06)	1.14 ± 1.36 ^a^ (0.62–1.65)	1.10 ± 1.32 ^a^ (0.60–1.61)	1.10 ± 1.42 ^a^ (0.56–1.64)	1.03 ± 1.84 ^a^ (0.33–1.74)	0.52 ± 0.91 ^b^ (0.17–0.86)	0.21 ± 0.62 ^c^ (−0.03–0.44)	0.14 ± 0.52 ^c^ (−0.06–0.33)	<0.001
	Reciprocating (R) ^§^	1.00 ± 1.56 ^a^ (0.34–1.66)	0.96 ± 1.62 ^a^ (0.29–1.70)	0.71 ± 1.30 ^a^ (0.16–1.26)	0.71 ± 1.16 ^a^ (0.22–1.20)	0.54 ± 1.02 ^a^ (0.11–0.97)	0.42 ± 0.97 ^a^ (0.01–0.83)	0.25 ± 0.74 ^a^ (−0.06–0.56)	0.21±0.59 ^a^ (−0.04–0.46)	0.08 ± 0.41 ^a^ (−0.09–0.26)	0.001
	*p* **	0.795	0.903	0.082	0.146	0.052	0.115	0.193	0.845	0.672	
	d	0.21	0.17	0.32	0.31	0.45	0.41	0.33	0.00	0.13	
NRS	Rotary (PTN) ^§^	1.48 ± 2.4 ^a^ (0.57–2.40)	1.45 ± 2.20 ^a^ (0.61–2.28)	1.41 ± 1.62 ^a^ (0.80–2.03)	1.31 ± 1.61 ^a^ (0.70–1.92)	1.45 ± 1.94 ^a^ (0.71–2.19)	1.07 ± 2.05 ^a^ (0.29–1.85)	0.41 ± 0.83 ^b^ (0.10–0.73)	0.28 ± 0.75 ^b^ (−0.01–0.56)	0.17 ± 0.54 ^b^ (−0.03–0.38)	<0.001
	Reciprocating (R) ^§^	1.13 ± 1.78 ^a^ (0.37–1.88)	1.33 ± 1.76 ^a^ (0.59–2.08)	0.92 ± 1.47 ^a^ (0.30–1.54)	0.96 ± 1.40^a^ (0.37–1.55)	0.88 ± 1.30 ^a^ (0.33–1.42)	0.42 ± 0.93 ^b^ (0.02–0.81)	0.46 ± 1.06 ^b^ (0.01–0.91)	0.25 ± 0.61 ^b^ (−0.01–0.51)	0.17 ± 0.48 ^b^ (−0.04–0.37)	0.002
	*p* **	0.983	0.770	0.094	0.304	0.188	0.228	0.845	0.841	0.845	
	d	0.17	0.06	0.32	0.23	0.35	0.41	−0.05	0.04	0	

* Intra-group comparison of all evaluated time intervals—Friedman test. ^§^ Comparison by pairs in each group—Wilcoxon test with Bonferroni correction (*p* < 0.006). ^a, b, c^ For each group, mean scores indicated by the superscript lowercase letters in same line are statistically different. The comparison was performed between pairs: 6–12 h; 12–24 h; 24 h–2 days; 2–3 days; 3–4 days; 4–5 days; 5–6 days; 6–7 days. ** Inter-group comparison of the same session for each evaluated time interval—Mann–Whitney test. d Effect size: d ≤ 0.20 (small), d from >0.20 to <0.80 (moderate), d ≥ 0.80 (large) [23]. PTN = ProTaper Next; R = Reciproc.

**Table 3 jcm-11-03816-t003:** Absolute and relative risk of postoperative pain (extracted from the NRS) and tenderness on biting (extracted from the yes-or-no question) after instrumentation (including all time intervals) for both groups.

		Group			*p*-Value	Absolute Risk (95% CI)	Relative Risk (95% CI)
Primary Outcome			No	Yes			
	Number of patients who experienced postoperative pain	Rotary (PTN) Reciprocating (R)	19 21	14 13	0.727 ^§^	0.10 (0.71–0.97) 0.41 (0.67–0.92)	1.33 (0.48–3.71)
Secondary Outcome	Number of patients who experienced tenderness on biting	Rotary (PTN) Reciprocating (R)	29 25	4 9	0.217 ^δ^	0.12 (0.05–0.27) 0.26 (0.67–0.92)	0.32 (0.08–1.32)

^§^—chi-squared of Pearson; ^δ^—Fisher’s exact test; CI—confidence interval; PTN—ProTaper Next; R—Reciproc.

**Table 4 jcm-11-03816-t004:** Non-adjusted and adjusted Poisson regression analyses for postoperative pain 12 h after root canal instrumentation according to the patient’s gender, oral hygiene status, tooth location and instrumentation technique.

	Postoperative Pain					
Independent variables	NO *n* (%)	YES *n* (%)	RR (95% CI)	*p*	RR-Adjusted (95% CI)	*p*
**Gender**				0.278		0.187
Male	14 (35)	13 (48.1)	1.40 (0.77–2.45)		1.46 (0.83–2.57)	
Female	26 (65)	14 (51.9)	1		1	
**Oral hygiene status**				0.921 0.015		0.424 0.012
Poor	2 (5.0)	3(11.1)	0.96 (0.43–2.16)		0.73 (0.34–1.57)	
Moderate	32 (80.0)	14 (51.9)	0.49 (0.27–0.87)		0.48 (0.27–0.85)	
Good	6 (15.0)	10 (37)	1		1	
**Tooth location**				0.041		0.085
Mandibular	23 (57.5)	23 (85.2)	2.63 (1.04–6.64)		2.33 (0.89–6.11)	
Maxillary	17 (42.5)	4 (14.8)	1		1	
**Instrumentation technique**				0.727	0.88 (0.49–1.58)	0.668
R	21 (52.5)	13 (48.1)	0.90 (0.50–1.62)			
PTN	19 (47.5)	14 (451.9)	1			

RR = Relative risk.

## Data Availability

The datasets used and/or analyzed during the current study are available from the corresponding author upon reasonable request.

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
