# Peer review of "Postoperative Pain Following Root Canal Instrumentation Using ProTaper Next or Reciproc in Asymptomatic Molars: A Randomized Controlled Single-Blind Clinical Trial"

_jcm, 2022, doi:10.3390/jcm11133816_

Round 1

Reviewer 1 Report

Thank you for your submission of this manuscript and your efforts to evaluate post-operative pain after endodontic instrumentation.

Although this study has positive aspects (single blind, randomized approach; standardized biomechanical instrumentation protocols; use of multiple pain scales) there are some significant issues with the described methodology. 

1. Was the periapical diagnosis of the teeth determined? Periapical testing and diagnoses are not reported. This is relevant for interpretation of the secondary outcome (pain on biting)

2. The description of the analgesic instructions/ reporting is unclear. 

a. were all patients eligible to take ibuprofen, according to their medical history? This should be clarified.

b. 400mg of ibuprofen were recommended to patients "if needed". Was the 400mg ibuprofen provided in a single tablet or multiple tablets? This information is relevant for the interpretation of results on mean number of pills taken.

3. It is unclear how the pain on biting data was collected- was this collected at a single time point or at multiple time points? This is not reflected in the results tables 1 or 2.

4. It is of concern that intracanal medication was not used in the procedure protocol between visits. This is not standard endodontic treatment protocol and the reasoning for this approach should be described/ justified.

Author Response

Dear Editors and Reviewers,

Thank you very much for the comments and suggestions about our paper. We are glad to have the opportunity to review our article. Please find below the answers for the reviewers` questions and the new version of our paper (the modifications in the text are highlighted in yellow).

We hope that this new version may reach the standards to be published in your journal.

Sincerely,

The authors

--------------------------------------------------------------------------------------

Response to Reviewers

Responses to Reviewer 1

Thank you for your submission of this manuscript and your efforts to evaluate post-operative pain after endodontic instrumentation. Although this study has positive aspects (single blind, randomized approach; standardized biomechanical instrumentation protocols; use of multiple pain scales) there are some significant issues with the described methodology.

  1. Was the periapical diagnosis of the teeth determined? Periapical testing and diagnoses are not reported. This is relevant for interpretation of the secondary outcome (pain on biting).

Thank you very much for noticing this fault. We included the periapical diagnosis in the sentence that reports the inclusion criteria (“normal apical tissues”). The tests that helped us to provide this diagnosis were cited on lines 97-98 (“probing, palpation and percussion”).

  1. The description of the analgesic instructions/ reporting is unclear.
  2. were all patients eligible to take ibuprofen, according to their medical history? This should be clarified.

Yes – the medical condition (non-contributory) for included patients was reported on lines 98-100. To make this recommendation clearer, the following sentence is now added on lines 152-153: “This recommendation was based on the non-contributory medical history of the included patients.”

  1. 400mg of ibuprofen were recommended to patients "if needed". Was the 400mg ibuprofen provided in a single tablet or multiple tablets? This information is relevant for the interpretation of results on mean number of pills taken.

Thank you for this helpful comment-

This information is now included into the text. Please see the new sentence in Lines 153-155. “As the chosen medication is commercially available under no-prescription, we chose to not provide patients with ibuprofen tables – and this also prevented overconsumption.”

Patients were given instructions to take the medication only if they felt mild-to-severe pain (400 mg – each 6 hours, while having pain).

  1. It is unclear how the pain on biting data was collected- was this collected at a single time point or at multiple time points? This is not reflected in the results tables 1 or 2.

The specific information about the ‘tenderness on biting’ data collection was reported on Lines 142-146. Patients answered a Yes/No question for each period evaluated that was included in the pain diary. Following your comment, we included in this new version details of how this should be assessed by the patients. Please see the new sentence on Lines 157-158. “Patients were instructed to position the latex device over the treated-tooth and bite it for approximately 3 s (up to three times)”.

  1. It is of concern that intracanal medication was not used in the procedure protocol between visits. This is not standard endodontic treatment protocol and the reasoning for this approach should be described/ justified.

Although this is not the standard protocol for a ‘real situation’ – the choice for not using intracanal medication in trials that evaluate canal shaping has been used to avoid any other variables that could (negatively or positively) interfere with patients` postoperative symptomology. In this current trial the preoperative pulp vitality (absence of contamination in the root canal system), and the short period between appointments (7 days) favored our choice for not using medication.

Reviewer 2 Report

This study analyses the effect of root canal instrumentation techniques rotary and reciprocating on the postoperative patient pain intensity and tenderness on biting. 

It is a well-written study that has merit. 

In the discussions section, the authors should discuss the possible different pain perception of women during their menstrual cycle.  It could be an interesting future study since this aspect has not been studied for endodontic treatment, only for periodontal and orthodontic pain.

Author Response

Responses to Reviewer 2

This study analyses the effect of root canal instrumentation techniques rotary and reciprocating on the postoperative patient pain intensity and tenderness on biting. It is a well-written study that has merit.

  • In the discussions section, the authors should discuss the possible different pain perception of women during their menstrual cycle. It could be an interesting future study since this aspect has not been studied for endodontic treatment, only for periodontal and orthodontic pain.

Thank you for reading and evaluating our paper. We considered your suggestion interesting for further trials – However, the collection of those data was outside the scope of this current study.

Round 2

Reviewer 1 Report

Thank you for taking the time to revise your submitted manuscript.

One comment has not been sufficiently addressed and appears to be a major methodological flaw in the study design.

Original Comment:

  1. 400mg of ibuprofen were recommended to patients "if needed". Was the 400mg ibuprofen provided in a single tablet or multiple tablets? This information is relevant for the interpretation of results on mean number of pills taken.

Response: 

This information is now included into the text. Please see the new sentence in Lines 153-155. “As the chosen medication is commercially available under no-prescription, we chose to not provide patients with ibuprofen tables – and this also prevented overconsumption.”

Patients were given instructions to take the medication only if they felt mild-to-severe pain (400 mg – each 6 hours, while having pain).

This response does not address the original comment. Patients were instructed to take 400 mg of ibuprofen, "as needed". If the analgesic was not prescribed and administered in a standard format to all patients, how can the authors be sure that "1 tablet" of ibuprofen taken by each patient contains the same number of mg? For this reasons, it appears that the mg of ibuprofen taken by each patient should have been recorded, rather than the number of tablets. This raises concern about the reliability of this reported outcome.

Author Response

We apologise for not being clear in our text.

Please see the new sentences highlighted in blue in this new version of the manuscript. We clarified in the text that patients did receive detailed recommendation about the dosage of the tables – and we also made clearer that the question in the pain diary contained the medication dosage. Other authors also used the same protocols of data collection in previous studies, such as:

Neelakantan & Sharma. Pain after single-visit root canal treatment with two single-file systems based on different kinematics--a prospective randomized multicenter clinical study. Clin Oral Investig. 2015 Dec;19(9):2211-7

and

Silva EJ, Menaged K, Ajuz N, Monteiro MR, Coutinho-Filho Tde S. Postoperative pain after foraminal enlargement in anterior teeth with necrosis and apical periodontitis: a prospective and randomized clinical trial. J Endod. 2013 Feb;39(2):173-6. doi: 10.1016/j.joen.2012.11.013.

The following parts (in blue) were the ones we modified into the text:

Postoperative Pain, Intake of Analgesic, and Tenderness on Biting Assessment

Each patient received a pain diary to record their pain at the following time-points: 6, 12 and 24 hours, and from day-2 to day-7 following the procedure. The pain diary consisted of two scales to record pain intensity and 2 questions, one about analgesic intake (number of 400 mg tablets), and the other about tenderness on biting (yes or no question). First, patients pointed a mark in a numerical rating scale (NRS) that was a ten-point scale (10 cm) measuring 0 for no pain, 1-2 for mild pain, 3-4 for moderate pain, 5-7 for considerable pain and 8-10 for severe pain. Second, patients pointed a mark in a visual analogue scale (VAS) that was a 10 cm horizontal line with only two scores at the ends: 0, for no pain, and 10 for severe pain. The distance from the patient`s mark to the 0 value was considered as a numerical value for the pain intensity. The patients were prescribed with medication following treatment; They were recommended to take ibuprofen only if they felt moderate-to-severe pain (400 mg – each 6 hours, while having pain) (25). This recommendation was based on the non-contributory medical history of the included patients. If the pain did not alleviate patients were instructed to contact the operator. For tenderness on biting, patients were given 3 x 2 cm latex devices to bite to be able to report the existence of this symptom. Patients were instructed to position the latex device over the treated-tooth and bite it for approximately 3 s (up to three times). Participants returned their pain diaries in their second appointment, scheduled for 7 days after the first visit.
